# AQP2: Mutations Associated with Congenital Nephrogenic Diabetes Insipidus and Regulation by Post-Translational Modifications and Protein-Protein Interactions

**DOI:** 10.3390/cells9102172

**Published:** 2020-09-26

**Authors:** Chao Gao, Paul J. Higgins, Wenzheng Zhang

**Affiliations:** Department of Regenerative & Cancer Cell Biology, Albany Medical College, Albany, NY 12208, USA; gaoc@amc.edu (C.G.); higginp@amc.edu (P.J.H.)

**Keywords:** nephrogenic diabetes insipidus, AVPR2, AQP2, mutation, trafficking, phosphorylation, ubiquitination, glycosylation, protein-protein interaction

## Abstract

As a rare hereditary disease, congenital nephrogenic diabetes insipidus (NDI) is clinically characterized by polyuria with hyposthenuria and polydipsia. NDI results from collecting duct principal cell hyporesponsiveness or insensitivity to the antidiuretic action of arginine vasopressin (AVP). The principal cell-specific water channel aquaporin-2 (AQP2) plays an essential role in water reabsorption along osmotic gradients. The capacity to accumulate AQP2 in the apical plasma membrane in response to decreased fluid volume or increased plasma osmolality is critically regulated by the antidiuretic hormone AVP and its receptor 2 (AVPR2). Mutations in *AVPR2* result in X-linked recessive NDI, the most common form of inherited NDI. Genetic defects in *AQP2* cause autosomal recessive or dominant NDI. In this review, we provide an updated overview of the genetic and molecular mechanisms of congenital NDI, with a focus on the potential disease-causing mutations in *AVPR2* and *AQP2*, the molecular defects in the AVPR2 and AQP2 mutants, post-translational modifications (i.e., phosphorylation, ubiquitination, and glycosylation) and various protein-protein interactions that regulate phosphorylation, ubiquitination, tetramerization, trafficking, stability, and degradation of AQP2.

## 1. Introduction

There are two major forms of diabetes insipidus: central diabetes insipidus (CDI) and nephrogenic diabetes insipidus (NDI). CDI is caused by the failure of the hypothalamic-pituitary axis to produce or release the appropriate physiologic amounts of vasopressin. NDI results from vasopressin hyporesponsiveness or insensitivity of the collecting duct principal cells, which are responsible for water reabsorption through coordinated actions of apical AQP2 and basolateral AQP3 and AQP4. In both CDI and NDI, the kidney is unable to concentrate urine, leading to the production of excess urine or polyuria. NDI can be further classified into acquired and congenital forms. Compared to congenital NDI, acquired NDI is significantly more prevalent and occurs in various pathophysiologic conditions. The primary cause of acquired NDI is chronic administration of lithium, which is frequently used to treat a common chronic psychiatric disease, bipolar disorder. NDI develops as a result of long-term lithium therapy, although obstruction of the urinary tract, hypercalcemia, hypokalemia, and protein malnutrition can also cause NDI [1]. Congenital NDI is a genetic disorder, which we will discuss in detail below. 

## 2. Signs and Symptoms

Symptoms often develop in patients with congenital NDI quickly after birth and the majority of newborns are diagnosed during the first year of life. The primary symptoms include polyuria, polydipsia, and nocturia. In addition, patients may exhibit a variety of other symptoms such as unexplained fever, poor feeding, constipation, diarrhea, vomiting, lethargy, irritability, and retching. Therefore, some infants may not be able to grow or gain weight at the normal rate. Adult patients gradually develop orthostatic hypotension, megacystis hydronephrosis, and hydroureter because of persistent polyuria (reviewed in [2]).

## 3. Genetic Basis of Congenital NDI 

There are two forms of congenital NDI: primary inherited NDI and secondary inherited NDI. The vast majority of the primary inherited NDI cases (90%) show an X-linked recessive mode of inheritance. This disorder results from mutations in the antidiuretic hormone arginine vasopressin receptor 2 gene (*AVPR2*) on the long arm of the X chromosome (Xq28) [3]. To date, a total of 287 potential disease-causing mutations in *AVPR2* have been described, with 177 (62%) being missense mutations (The Human Gene Mutation Database at the Institute of Medical Genetics in Cardiff; http://www.hgmd.cf.ac.uk). The second major type involves small deletions, with an incidence of approximately 18%. Splicing mutations, small insertions, and gross deletions have been also reported, although they occur less frequently. 

About 10% and 1% of cases of the primary inherited NDI exhibit an autosomal recessive or dominant mode of inheritance, respectively. In both scenarios, the disorder is associated with defects in *AQP2* on the long arm of chromosome 12 (12q13). The first such NDI patient described was a male, who was determined to be a compound heterozygote for two *AQP2* missense mutations (R187C and S217P) [4]. Since then, at least 70 putative disease-causing *AQP2* mutations have been reported in 74 families (Figure 1 and Table 1). Among them are 54 missense mutations, 4 splicing mutations, 9 small deletions, 1 gross mutation, and 2 small insertions (The Human Gene Mutation Database at the Institute of Medical Genetics in Cardiff; http://www.hgmd.cf.ac.uk). In brief, inactivation of *AVPR2* or *AQP2* is the genetic basis of congenital NDI. 

The secondary inherited form refers to NDI as a complication associated with inherited human diseases, such as Bartter syndrome, apparent mineralocorticoid excess, Renal Fanconi syndrome, cystic kidney disorders, familial hypomagnesemia with hypercalciuria and nephrocalcinosis, and distal renal tubular acidosis (reviewed in [39]).

## 4. The AVP-AVPR2-AQP2 Signaling Pathway and Beyond

The general mechanisms involved in the AVP-AVPR2-AQP2 signaling pathway have been reviewed in detail elsewhere [40,41], only a succinct overview is given here. In response to decreased plasma volume or increased plasma osmolality, AVP is generated in the hypothalamus and secreted from the pituitary gland. AVPR2 is the cognate G protein-coupled receptor for AVP. In the principal cells of the collecting duct, binding of AVP to AVPR2 triggers a signaling cascade including involvement of a number of kinases, changes in the activities of phosphatases and other enzymes, and reorganization of the cytoskeleton. Collectively, these signaling events facilitate AQP2 routing to the apical plasma membrane and inhibit endocytosis-mediated AQP2 internalization. Consequently, there is an accumulation of AQP2 in the apical plasma membrane permitting water to move down its osmotic gradient through the membrane into the interstitium and subsequently into the circulation. AVP stimulation of AVPR2 also activates the adenylyl cyclases AC3 and AC6, promoting conversion of ATP to cyclic adenosine monophosphate (cAMP) mobilizing cAMP-dependent kinases such as PKA to function in AQP2 trafficking. AQP2 translocation can also take place in a cAMP-independent manner (reviewed in [2]).

The short-term effect of AVP on AQP2 is the movement of AQP2 from the intracellular vesicles to the apical membrane. A sustained increase of circulating AVP for 24 h or more can induce a long-term effect on water reabsorption by increasing the abundance of AQP2 [42,43]. This increase is considered as a consequence of enhanced AQP2 transcription [42,44]. 

Apart from the AVP-AVPR2-AQP2 signaling pathway, AVP affects other targets including the urea transporter UT-A1 [45], and the epithelial sodium channel [46]. Hence, AVP selectively increases the permeabilities of the apical membrane to water, urea, and sodium.

AQP2 is also controlled by multiple endogenous signaling molecules/hormones. They modulate AQP2 trafficking primarily through G protein-coupled receptors. Important among them are purines, calcitonin, secretin, glucagon, angiotensin II, serotonin, and prostaglandins (reviewed in [47]). Various alternative receptors also appear to impact AQP2 as well (reviewed in [48]). These include the farnesoid X receptor [49], the peroxisome proliferator-activated receptor-γ [50], angiotensin II receptor type I [51], the bile acid receptor TGR5 [49], the glucocorticoid and mineralocorticoid receptors [52], the estrogen receptor-α [53], and the liver X receptor-β [48].

## 5. Molecular Defects of AVPR2 Mutants 

Genetic defects in *AVPR2* may result in: (1) improperly processed or unstable mRNA like promoter alterations, exon skipping, aberrant splicing, frame-shift or non-sense mutations, which result in truncated proteins, (2) decreased binding capacity of the receptor to AVP, or (3) mis-folding of the receptor that is retained intracellularly and incapable of routing to the basolateral membrane to bind AVP (reviewed in [54,55]). The vast majority of the *AVPR2* mutations lead to mis-folding of the receptor [56]. The mis-folded receptors concentrate in the Golgi apparatus and are subsequently degraded by the ubiquitin-proteasome pathway, conferring loss-of-function phenotypes on AVPR2 [57]. Rescuing the normal function of these mis-folded proteins, therefore, may provide new therapeutically useful strategies for the management of NDI patients. The key step in any such approach is to facilitate the freeing of the trapped receptor from the intracellular compartments. Nonpeptide inhibitors could be explored to rescue the mis-folded AVPR2 variants by acting as pharmacologic chaperons rather than endocytosis antagonists [58]. Indeed, administration of the vasopressin inhibitor SR49059 significantly ameliorated the symptoms of polyuria and polydipsia without affecting other renal physiological parameters such as creatinine excretion and electrolyte measurements [59]. These findings support the concept that pharmacologic chaperons might have clinical utility in the management of some forms of congenital NDI.

## 6. Molecular Defects of Recessive AQP2 Mutants

Patients with congenital autosomal recessive NDI are homozygous for a mutation in *AQP2* or carry two different *AQP2* mutations. Currently, all AQP2 mutations causing recessive NDI are located throughout the six transmembrane domains and five connecting loops of AQP2. With a few exceptions that possess a residual water transport capacity, these mutants are non-functional [9,12,60] and are retained in the endoplasmic reticulum (ER) as evidenced by their scattered ‘reticular’ expression pattern, by their unglycosylated (29 kDa) or high mannose glycosylated (32 kDa) variants, and by their decreased stability in comparison with WT AQP2 [60,61,62]. Consistent with the recessive nature of inheritance and possibly because of their mis-folding, these AQP2 mutants exist as monomers and are incapable of heteroligomerizing with and impeding the routing and maturation of WT AQP2 in its pathway to the apical membrane. Typical examples showing all these features are the AQP2-R187C and AQP2-A190T mutants [38,60,63,64]. R187 and A190 reside in one of the two most highly conserved fragments of the major intrinsic protein family. The signature motif consisting of three invariant amino acids (Asn-Pro-Ala) in this region is positioned just proximal to the mutated R187 and A190. Both R187 and A190 themselves are also strongly conserved in the major intrinsic protein family. 

## 7. Molecular Defects of Dominant AQP2 Mutants

Unlike recessive AQP2 mutants, the dominant AQP2 mutants are functional water channels. However, they are mis-routed to other subcellular compartments rather than to the apical membrane [33,36,37,38]. Because these AQP2 dominant mutants are correctly folded in the ER, they interact with WT-AQP2 to form heterotetramers and alter intracellular trafficking of the WT and mutant AQP2 complexes [65]. Consequently, the apical membrane of the renal collecting duct principal cells lacks sufficient levels of WT-AQP2, leading to dominant NDI. In contrast to the recessive mutants, all AQP2 mutations identified in dominant NDI are detected in the carboxyl terminus of AQP2, highlighting the crucial role of this domain in regulating the AQP2 sorting. AQP2-P262L, nevertheless, is a unique mutation, which was identified in two families with recessive NDI [64]. These patients are compound heterozygotes for AQP2-P262L together with AQP2-A190T or AQP2-R187C. Functional analyses in oocytes revealed that AQP2-P262L was a properly folded and functional aquaporin. AQP2-P262L localized to intracellular vesicles and was not retained in the ER when expressed in polarized cells. Upon co-expression, AQP2-P262L formed heteroteramers with WT-AQP2, but not with AQP2-R187C, leading to a rescued apical membrane localization of AQP2-P262L [64]. Hence, AQP2-P262L exhibited a cellular phenotype clearly distinct from other AQP2 mutants in recessive NDI. Instead, and in line with the molecular location of the mutation, it showed all the characteristics observed for AQP2 mutants in dominant NDI. Nevertheless, the mis-sorting of AQP2-P262L, unlike other AQP2 dominant mutants, is overruled by apical sorting of WT-AQP2, resulting in the carriers of the P262L mutation being asymptomatic as are those of AQP2 recessive mutations [64].

## 8. Regulation of AQP2 by Post-Translational Modifications

Post-translational modifications play a crucial role in signaling transduction, protein maturation and folding by altering the cellular distribution, stability, function, and binding partners of their substrate proteins. Recent studies suggest that phosphorylation, ubiquitination, and glycosylation are major post-translational modifications that regulate AQP2 function, routing, stability, and protein–protein interactions.

### 8.1. Phosphorylation

Bioinformatic analysis identified a number of putative phosphorylation sites in AQP2 for various kinases including protein kinases A (PKA) and casein kinase II [66], with the *C*-terminal S256 residue perhaps being the best characterized. As discussed above, binding of AVP to AVPR2 triggers a signaling cascade that leads to increased cAMP levels. One of the cAMP effectors is PKA. The PKA holoenzyme is a heterotetramer comprised of two inactive catalytic subunits and two regulatory subunits. Each regulatory domain binds one catalytic subunit. Upon interaction of cAMP with the PKA regulatory elements, the catalytic subunits are released from the heterotetramer, allowing them to function as active serine-threonine kinases, phosphorylating several substrates including AQP2. PKA-catalyzed AQP2 phosphorylation at S256 increases the rate of exocytosis [66]. Altering cAMP abundance and/or PKA enzymatic activity with calcitonin or prostaglandin E2 changes both AQP2 phosphorylation and trafficking [67,68]. Nevertheless, the AQP2 S256 residue can be phosphorylated and, thereby, regulated in response to vasopressin even in PKA-null cells, indicating that AVPR2-mediated AVP signaling in collecting duct cells is not entirely PKA-dependent [69].

Casein kinase II also phosphorylates S256, which is necessary for transition of AQP2 through the Golgi [70]. Vasopressin or forskolin was unable to stimulate translocation of the AQP2-S256A mutant when expressed in LLC-PK1 cells [71,72]. Since the AQP2-S256L mutation also prevents phosphorylation at S256 and the subsequent accumulation of AQP2 on the apical membrane of the collecting duct principal cells, mice homozygous for this mutation had severe urine concentration defects and developed congenital progressive hydronephrosis [73]. These studies illustrate a critical role of S256 phosphorylation in the apical targeting of AQP2 and thus water transport.

Large-scale phospho-proteomic analysis later revealed that AQP2 S261, S264, and S269 are additional AVP-mediated phosphorylation targets [74,75]. Phosphorylation at these sites also impacts AQP2 trafficking. S261 phosphorylation may stabilize ubiquitinated AQP2 and intracellular localization, since it is primarily found in the intracellular vesicles after ubiquitination and endocytosis and S261 phosphorylation is reduced in response to AVP treatment [76]. S269 phosphorylation has been detected exclusively on the plasma membrane [75], however, suggesting a role in promoting AQP2 plasma membrane targeting and/or in inhibiting AQP2 endocytosis [77]. Phosphorylation at S256 appears unnecessary for AQP2 recycling because substitution of S256 into alanine has little effect since AQP2-S256A recycles rapidly and constitutively. Nevertheless, S256 phosphorylation is essential for subsequent phosphorylation of other *C*-terminal serines [75] since phosphorylated forms of AQP2 at S264 and S269 are not detected in cells expressing the AQP2 S256 mutant or in kidneys from AQP2 S256 mutant mice [75]. 

### 8.2. Ubiquitination

The ubiquitin proteasome and lysosomal proteolysis pathways are two primary mechanisms for protein degradation in mammalian cells. Ubiquitination is a complex post-translational modification in which ubiquitin is covalently added to a target protein through a system consisting of E1 activation, E2 conjugation, and E3 ligation enzymes. Ubiquitination facilitates protein endocytosis and lysosomal degradation as well as impacting the interaction of the target protein with its binding partners. AQP2 possesses three potential ubiquitination sites (K228, K238, and K270) [78]. Mutagenesis analyses revealed, however, that ubiquitination occurs only at K270 whereby a K63-linked chain comprising generally two to three ubiquitin moieties is attached [78]. This modification at the plasma membrane enhances AQP2 endocytosis, routing to intraendosomal vesicles and subsequent degradation [78]. Cullin-5 is a vasopressin-activated calcium-mobilizing receptor and a member of the Cullin family of scaffold proteins of the E3 complex. It may be involved in the attachment of ubiquitin to AQP2, leading to ubiquitination, internalization, and lysosomal and/or proteosomal degradation of AQP2 after 1-desamino-8-d-arginine vasopressin withdrawal [79]. Analyses of several large-scale transcriptomic and proteomic datasets defined a panel of E3 ligases that are most likely to form complexes with AQP2, with NEDD4 and NEDD4L being the most prominent [80], although there are other required effectors. Indeed, while NEDD4 and NEDD4L mediate ubiquitination and degradation of AQP2, NDFIP is required to link NEDD4 and NEDD4L to AQP2 [81].

It appears that AQP2 phosphorylation at S256, S261, S264, S269 and ubiquitination at K270 are coordinately regulated to fine tune AQP2 intracellular distribution. Phosphorylation usually takes place as a priming event for ubiquitination which, in turn, affects protein phosphorylation by regulating kinase activities [82]. Moreover, AQP2 phosphorylation can nullify the dominant endocytic signal of K63-linked polyubiquitination. Specifically, phosphorylation at S269 and ubiquitination at K270 of AQP2 can occur simultaneously, with elevated S269 phosphorylation and lowered AQP2 endocytosis occurring in the presence of the maximal K270 polyubiquitination levels [83]. Hence, AQP2 phosphorylation is apparently able to counterbalance polyubiquitination and governs its final localization. 

### 8.3. Glycosylation

Glycosylation is the enzymatic process in which oligosaccharides (also referred as to glycans) are attached to proteins. *N*-linked glycans are almost always attached to the nitrogen atom of an asparagine (Asn) side chain. This particular Asn appears as a part of the Asn-X-Ser/Thr consensus sequence, where X is any amino acid excluding proline [84]. The sequence Asn123-Ser124-Thr125 in the second extracellular loop of AQP2 is a consensus *N*-linked glycosylation site.

Glycosylation induces a complex effect on AQP2 function. In an AQP2-transfected Madin–Darby canine kidney cell line (clone WT10), 34% of the AQP2 molecules were glycosylated, which was reduced to 2% after treatment with the glycosylation inhibitor tunicamycin. Moreover, in tunicamycin-treated WT10 cells, all the AQP2 in the apical membrane was unglycosylated, whereas in untreated cells 30% of AQP2 in the apical membrane was glycosylated. These observations suggest that glycosylation is not involved in the routing of AQP2 in Madin–Darby canine kidney cells [85]. Analyses of the AQP2 N123Q mutant, which carries the disrupted *N*-linked glycosylation consensus site, revealed that this mutation still allowed for generation of the nonglycosylated 29-kDa precursor, but neither the 31-kDa nor the mature glycosylated 42-46-kDa variants. The mutant protein, however, had a shorter half-life, compared to WT-AQP2 [86]. A shorter half-life was also observed for the AQP2-S256D and AQP2-E258K mutants [86], although their glycosylation was similar to WT-AQP2 [86]. These findings indicate that glycosylation per se is not crucial for AQP2 folding and that AQP2 folding is independent of the primary calnexin/calreticulin ER quality control mechanisms [86,87]. The AQP2-T125M mutant identified in patients with recessive NDI [9,20], nevertheless, is not glycosylated due to the loss of the consensus N-linked glycosylation sequence. This mutant is not localized to the plasma membrane [9], implying that it is not correctly folded and retained in the ER, or because of other unknown mechanisms, does not translocate to the cell surface. Further analyses of the AQP2-N123Q revealed that glycosylation is neither required for tetramerization in the ER nor for trafficking from the ER to the Golgi complex. Addition of the N-linked glycan, instead, is critical for exit from the Golgi complex and routing of AQP2 to the plasma membrane [86], conflicting with what was reported in WT10 cells [85].

## 9. Regulation of AQP2 by Protein–Protein Interactions

Multiple proteins interact with AQP2 and regulate AQP2 degradation, post-translational modification, and trafficking (Figure 2 and Table 2). 

One of the AQP2-binding proteins is lysosomal trafficking regulator interacting protein 5 (LIP5) [88]. LIP5 functions in multivesicular body formation. The complex formation between LIP5 and the AQP2 *C*-terminal tail promotes AQP2 lysosomal degradation and occurs regardless of the state of phosphorylation at S256 and ubiquitination at K270 [88]. This interaction was identified by yeast two-hybrid and confirmed by glutathione S-transferase pull-down, co-immunoprecipitation, and colocalization assays. Knockdown of LIP5 in mpkCCD cells attenuated phorbol ester-induced AQP2 degradation [88]. LIP5, therefore, appears important in the regulation of AQP2 degradation, possibly by inhibiting the formation of late endosomes [88].

Dynamic intracellular mobilization of AQP2 depends on its specific interactions with the components of the trafficking machinery and the actin cytoskeleton. These include Caveolin-1 [89], the myelin and lymphocyte-associated protein (MAL) [90], the signal-induced proliferation-associated gene-1 (SPA-1) [91], the 70-kDa heat shock proteins hsc70 and hsp70 [93], A-kinase anchoring protein 220 (AKAP220) [95], 14-3-3 [102], Ezrin [96], and AQP5 [97]. Each of these AQP2-intercating proteins is co-expressed and colocalized with AQP2 in the collecting duct cells [89,90,91,93,95,96,97,102].

AQP2-Caveolin-1 interaction is believed to regulate the internalization of AQP2 through a caveolin-1-dependent pathway in MDCK cells [89]. MAL is a detergent-resistant membrane-associated protein involved in apical sorting events and AQP2 phosphorylation at S256 appears to enhance its binding affinity with MAL. While MAL does not affect apical delivery of AQP2 or its detergent-resistant association with membrane in renal epithelial cells, it promotes the S256 phosphorylation and apical localization of AQP2 by inhibiting its internalization [90]. 

SPA-1, a PDZ-domain containing GTPase-activating protein for Rap1 was identified in a biochemical search for AQP2-interacting proteins [91]. The physiological relevance and significance of this interaction is highlighted by the coincided distribution of SPA-1 with that of AQP2 in renal collecting ducts, the concomitant co-relocation by hydration status, and the impaired AQP2 trafficking to the cell surface membrane in MDCK cells expressing a SPA-1 mutant lacking Rap1 GTPase-activating protein activity or the constitutively active mutant of Rap1 [91]. Moreover, AQP2 apical localization was reduced in SPA-1-deficient mice. These studies illustrate that SPA-1 interacts with AQP2 and promotes at least partially AQP2 trafficking [91].

The 70-kDa heat shock proteins were uncovered as proteins interacting with the AQP2 *C*-terminus in a yeast two-hybrid screening [93]. The interaction was found to be direct, partially inhibited by ATP, and critically affected by the Ser-256 residue in the AQP2 *C* terminus. Vasopressin enhances their interaction, as evidenced by increased binding in immunoprecipitation assays and by an increased co-localization of the two proteins on the apical membrane of principal cells in rat collecting ducts. Hsc70 knockdown increased membrane accumulation of AQP2 and reduced endocytosis of rhodamine-transferrin. Hence, the interaction influences AQP2 trafficking [93].

Structurally diverse A-Kinase Anchoring Proteins (AKAPs) share a common function: acting as scaffold proteins binding to PKA and other signaling proteins and physically recruiting these protein complexes to distinct cellular locations. This enables regulation of specific substrates through phosphorylation by PKA and dephosphorylation by phosphatases. In response to dehydration, PKA phosphorylates AQP2 at S256 to promote its apical membrane sorting. AKAP220 interacts with AQP2 in yeast two-hybrid assays and co-localizes with AQP2 in the cytosol of the inner medullary collecting ducts [95]. AKAP220 co-expression increases forskolin-stimulated phosphorylation of AQP2 expressed in transiently transfected COS cells. Based on these results, AKAP220 is thought to enhance AQP2 trafficking by recruiting PKA to phosphorylate AQP2 [95]. The low molecular weight compound 3,3′-diamino-4,4′-dihydroxydiphenylmethane (FMP-API-1) and its derivatives are disruptors of the AKAP-PKA complex. These drugs elevated AQP2 membrane activity to the same extent as vasopressin in mpkCCD cells and robustly activated AQP2 in an AVPR2-inhibited NDI mouse model [103]. Direct activation of PKA by dissociating AKAP led to the phosphorylation of a very specific group of proteins compared to G protein-coupled receptor agonists or adenylyl cyclase activators such as forskolin, which induced phosphorylation of many more PKA substrates (reviewed in [47]). FMP-API-1/27 appears to be a renal-specific AKAP-PKA disruptor, which acted mainly in collecting ducts without phosphorylating most of the PKA substrates in the whole kidney and heart [103]. This specificity would lower the potential adverse side effects in other cells and organs. Hence, FMP-API-1/27 may represent a promising lead compound for the treatment of congenital NDI caused by *AVPR2* mutations [103].

Proteins that differentially bind to phosphorylated versus non-phosphorylated peptides of the AQP2 *C*-terminus were identified by mass spectroscopy and validated by co-immunoprecipitation [94]. This approach not only confirmed previously documented hsc70 and hsp70 as AQP2 binding partners, but also revealed a higher affinity of these proteins to the nonphosphorylated form rather than to the S256-phosphorylated AQP2. Contrarily, the interaction of hsp70-5 (BiP/grp78), another heat shock protein, with phosphorylated AQP2 was found to be stronger when compared to the nonphosphorylated protein [94]. In aggregate, these data support the concept that S256 phosphorylation governs AQP2 trafficking by modulating interactions of hsp70 family members with the AQP2 *C*-terminus [94].

Phosphorylation at S256 and other sites in the AQP2 *C*-terminus also modulates AQP2 interaction with 14-3-3ζ and -θ [102]. 14-3-3ζ silencing increased AQP2 ubiquitination, reduced AQP2 protein half-life, and decreased AQP2 levels while 14-3-3θ knockdown had an opposite effect [102]. The binding of 14-3-3 proteins to AQP2, therefore, apparently impacts AQP2 phosphorylation, trafficking, ubiquitination, and degradation [102]. 

Ezrin is an actin-binding protein that also interacts with AQP2 [96]. Knockdown of Ezrin elevated AQP2 membrane accumulation and decreased AQP2 endocytosis. Direct binding of Ezrin to AQP2 promotes AQP2 endocytosis, linking AQP2 trafficking to the dynamic actin cytoskeletal network [96]. 

We reported that AQP5 interacts with Aqp2 in multiple assays [97]. AQP2 and AQP5 are close homologs and possess 66% sequence identity. AQP5 is normally synthesized in the eyes, salivary glands, lung, and sweat glands [104,105,106]. Impaired AQP5 trafficking in the lacrimal gland results in Sjögren’s syndrome, featured by dry eye and mouth [107]. While Aqp5 is undetectable in the normal mouse kidney [108], it is readily detectable and co-localizes with Aqp2 in *Dot1l^AC^* mice in which the histone H3 K79 methyltransferase *Dot1l* is specifically inactivated in the epithelial cells of the connecting tubule/collecting duct. Overexpression of AQP5 reduced AQP2 cell surface expression. AQP5 is also expressed and co-localizes with AQP2 at the perinuclear region of the collecting duct cells in patients with diabetic nephropathy [97]. These discoveries demonstrate that AQP5 may regulate AQP2 trafficking, possibly by forming heterotetramers with AQP2, and that the impaired apical localization of AQP2 because of abnormal AQP5 expression may contribute to the deterioration of glucosuria-induced polyuria in diabetic patients as well as to the development of hypoosmotic polyuria in *Dot1l^AC^* mice [97,109]. 

Most recently, an AQP2 interactome listing 139 AQP2-interacting proteins was reported (https://hpcwebapps.cit.nih.gov/ESBL/Database/AQP2-interactome/) [110], including the 70-kDa heat shock proteins and 14-3-3 proteins discussed previously. This interactome offers an overall picture of a dynamic biological process in which AQP2 is synthesized in the rough ER, matures via the Golgi apparatus, transported to endosomes that move into or out of the plasma membrane, and regulated in the plasma membrane [110]. 

## 10. Summary and Perspectives

AQP2-mediated water reabsorption in the collecting duct principal cells is tightly controlled by the molecular action of AVP through its receptor AVPR2. AVP binding to AVPR2 triggers a signaling cascade, leading to increased apical localization of AQP2 and, thus, increased osmotic water permeability of the apical membrane in collecting duct principal cells. To date, a total of 287 (in *AVPR2*) and 70 (in *AQP2*) potential disease-causing mutations have been identified in patients with hereditary NDI. These mutations cause or are suggested to cause mis-folding, mis-routing, impaired binding to AVP (for AVPR2 mutants), and/or impaired function as a water channel (for AQP2 mutants). Phosphorylation, ubiquitination, and glycosylation are the major post-translational modifications that regulate AQP2 trafficking and degradation. Numerous AQP2 binding partners promote or inhibit AQP2 post-translational modifications, translocation to the plasma membrane, and/or degradation. AQP2 appears to be a “model membrane protein” for elucidating protein trafficking in epithelial cells, and an excellent candidate to evaluate the role of post-translational controls on protein function and the highly interactive hormone-regulated signaling networks that coordinate exocytic and endocytic processes. As more data regarding the sophisticated signaling and regulatory mechanisms regulating AQP2 trafficking is gained, novel potential therapeutic strategies for NDI will likely emerge [47]. Successful development of these strategies may hold promise for future management of the disease.

## Figures and Tables

**Figure 1 cells-09-02172-f001:**
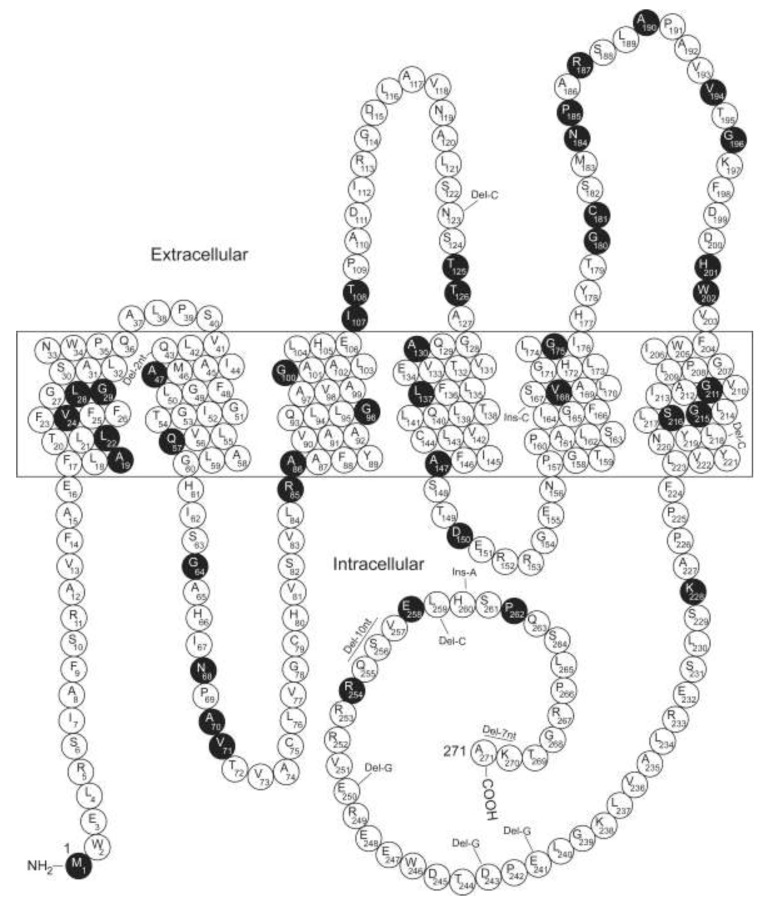
Diagram of the AQP2 protein and the locations of the affected residues by the putative disease-causing AQP2 mutations. AQP2 is depicted with six transmembrane domains. The extracellular, transmembrane and cytoplasmic domains are represented as reported [4]. Solid symbols indicate the affected residues because of the mutations (Table 1).

**Figure 2 cells-09-02172-f002:**
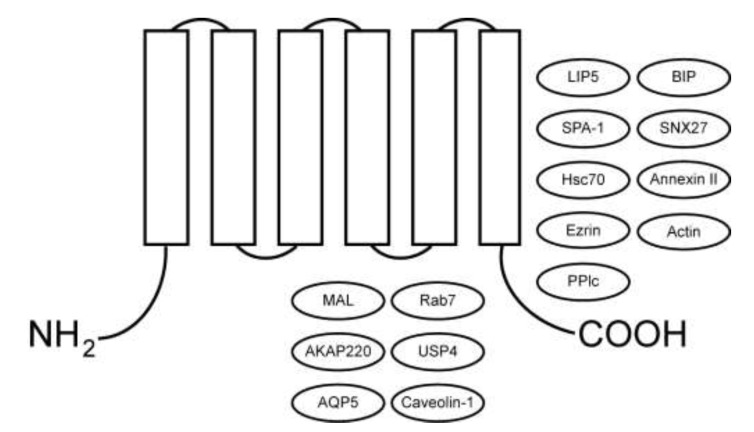
AQP2-interacting proteins. AQP2 binding partners are indicated, as well as the location for the interaction in those cases where it has been mapped (Table 2).

**Table 1 cells-09-02172-t001:** Potential disease-causing mutations in *AQP2.*

	No. of Family	Accession Number	Name of Mutation	Domain	Nucleotide Change	Predicted Change	Ref
Mis-sense	1	CM086773	M1I	NH2	ATG-to-ATT	Met-to-Ile	[5]
	1	CM137423	A19V	TMI	GCC-to-GTC	Ala-to-Val	[6]
	1	CM970097	L22V	TMI	CTC-to-GTC	Leu-to-Val	[7]
	1	CM106380	V24A	TMI	GTC-to-GCC	Val-to-Ala	[8]
	1	CM024085	L28P	TMI	CTC-to-CCC	Leu-to-Pro	[9]
	1	CM086774	G29S	TM1	GGC-to-AGC	Gly-to-Ser	[5]
	2	CM024086	A47V	TMII	GCG-to-GTG	Ala-to-Val	[9]
	2	CM021246	Q57P	TMII	CAG-to-CCG	Glu-to-Pro	[10]
	1	CM940082	G64R	CII	GGG-to-AGG	Gly-to-Arg	[11]
	1	CM970098	N68S	CII	AAC-to-AGC	Asn-to-Ser	[12]
	1	CM056523	A70D	CII	GCC-to-GAC	Ala-to-Asp	[13]
	2	CM950080	V71M	CII	GTG-to-ATG	Val-to-Met	[14]
	1	CM153550	A86V	TMIII	GCC-GTC	Ala-to-Val	[15]
	1	CM138317	G96Q	TMIII	GGG-GAG	Gly-to-Glu	[16]
	2	CM021247	G100V	TMIII	GGA-to-GTA	Gly-to-Val	[10]
	1	CM062427	G100R	TMIII	GGA-to-AGA	Gly-to-Arg	[17]
	1	CM068766	I107N	EII	ATC-to-AAC	Ile-to-Asn	[18]
	1	CM146787	T108M	EII	ACG-ATG	Thr-to Met	[19]
	1	CM980100	T125M	EII	ACG-to-ATG	Thr-to-Met	[20]
	1	CM970100	T126M	EII	ACG-to-ATG	Thr-to-Met	[12]
	1	CM1411333	A130V	TMIV	GCG-to-GTG	Ala-to-Val	[21]
	1	CM1210558	L137P	TMIV	CTG-to-CCG	Leu-to-Pro	[22]
	1	CM970101	A147T	TMIV	GCC-to-ACC	Ala-to-Thr	[12]
	2	CM071560	D150E	ICII	GAT-to-GAA	Asp-to-Glu	[23]
	1	CM973106	V168M	TMV	GTG-to-ATG	Val-to-Met	[24]
	1	CM980101	G175R	TMV	GGG-to-AGG	Gly-to-Arg	[20]
	1	CM062428	G180S	EIII	GGC-to-AGC	Gly-to-Ser	[17]
	1	CM970102	C181W	EIII	TGC-to-TGG	Cys-to-Trp	[7]
	1	CM1411334	N184H	EIII	AAT-to-CAT	Asn-to-His	[21]
	1	CM950081	P185A	EIII	CCT-to-GCT	Pro-to-Ala	[14]
	3	CM940083	R187C	EIII	CGC-to-TGC	Arg-to-Cys	[11]
	1	CM056522	R187H	EIII	CGC-to-CAC	Arg-to-His	[13]
	1	CM950082	A190T	EIII	GCT-to-ACT	Ala-to-Thr	[14]
	1	CM024087	V194I	EIII	GTC-to-ATC	Val-to-Ile	[9]
	1	CM087015	G196D	EIII	GGC-to-GAC	Gly-to-Asp	[25]
	1	CM120677	H201Y	EIII	CAC-to-TAC	His-to-Tyr	[26]
	1	CM960073	W202C	EIII	TGG-to-TGT	Trp-to-Cys	[27]
	1	CM120678	G211R	TMVI	GGC-to-CGC	Gly-to-Arg	[26]
	1	CM156813	G215S	TMVI	GGC-to-AGC	Gly-to-Ser	[28]
	1	CM071561	G215C	TMVI	GGC-to-TGC	Gly-to-Cys	[23]
	2	CM940084	S216P	TMVI	TCC-to-CCC	Ser-to-Pro	[11]
	1	CM099886	S216F	TMVI	TCC-to-TTC	Ser-to-Phe	[29]
	1	CM106381	K228E	CIV	AAG-to-GAG	Lys-to-Glu	[8]
	1	CM096039	R254Q	CIV	CGG-to-CAG	Arg-to-Gln	[30]
	1	CM056296	R254L	CIV	CGG-to-CTG	Arg-to-Leu	[31]
	1	CM1514252	R254W	CIV	CGG-to-TGG	Arg-to-Trp	[32]
	1	CM980102	E258K	CIV	GAG-to-AAG	Glu-to-Lys	[33]
	2	CM950083	P262L	CIV	CCG-to-CTG	Pro-to-Leu	[14]
Non-sense	2	CM972978	R85X	CII	CGA-TGA	Arg-to-Stop	[24]
	1	CM970099	G100X	TMIII	GGA-to-TGA	Gly-to-Stop	[34]
Frame-shift	1	CD034849	127–128del	TMII	2bp deletion	Post-elongation	[35]
	1	CD941595	369delC	EII	1bp deletion	Stop at Codon 131	[11]
	1	CD024169	652delC	TMVI	1bp deletion	Post-elongation	[9]
	1	CD014628	721delG	CIV	1 bp deletion	Post-elongation	[36]
	1	CD024654	727delG	CIV	1 bp deletion	Post-elongation	[37]
	1	CD137424	750delG	CIV	1 bp deletion	Post-elongation	[6]
	1	CD014629	763–772del	CIV	10 bp deletion	Post-elongation	[36]
	1	CD137425	775delC	CIV	1 bp deletion	Post-elongation	[6]
	1	CI156810	501–502insC	CIV	1 bp insertion	Post-elongation	[28]
	1	CI034768	779–780insA	CIV	1 bp insertion	Post-elongation	[38]
	1	CD983834	812–818del	CIV	7 bp deletion	Post-elongation	[36]
Splicing site	1	CS1213334	IVS2-1G>A	NA	G-to-A	NA	[14]
	1	CS024161	IVS3+1G>A	NA	G-to-A	NA	[9]
	1	CS034835	IVS3-1G>A	NA	G-to-A	NA	[35]

Note: All mutations listed in http://www.hgmd.cf.ac.uk as of 09/20/2020 are presented.

**Table 2 cells-09-02172-t002:** AQP2-interacting proteins.

Target	Interacting Protein	Proposed Function	Method Applied					Ref
			YTH	PD	IP	IHC	MS	
hAQP2	LIP5	Lysosomal degradation	X^a^	X^a^	X	X		[88]
hAQP2	Caveolin-1	AQP2 internalization			X	X		[89]
hAQP2	MAL	Increases apical surface expression			X	X		[90]
rAQP2	SPA-1	Regulates trafficking to the apical membrane		X^a^	X	X	X	[91,92]
rAQP2	Hsc70	Co-localized in the apical membrane: involved in trafficking	X^ac^		X	X	X	[93,94]
hAQP2	AKAP220	Phosphorylation of AQP2 triggering trafficking	X^d^			X^b^		[95]
rAQP2	Ezrin	Endocytosis		X^d^	X^d^	X		[96]
hAQP2	hAQP5	Impairing AQP2 membrane localization			X	X		[97]
rAQP2	Annexin II	Phosphorylation dependent binding to the *C*-terminus			X		X	[94]
rAQP2	PP1c	Phosphorylation dependent binding to the *C*-terminus		X^d^	X		X	[94]
rAQP2	Bip	Phosphorylation dependent binding to the *C*-terminus		X^d^	X	X		[94]
rAQP2	Actin	Trafficking			X		X	[94,98]
mAQP2	Rab7	Trafficking			X	X	X	[94,99]
mAQP2	USP4	Deubiquitination			X	X		[100]
rAQP2	SNX27	Lysosomal degradation		X	X	X		[101]

Note: a: using AQP2 *C*-terminus; b: using rat kidney; c: using human kidney cDNA library; d: using full-length AQP2. YTH: yeast two-hybrid; PD: glutathione S-transferase pull-down; IP: immunoprecipitation; IHC: immunohistochemistry; MS: mass spectroscopy.

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
