# Peer review of "AQP2: Mutations Associated with Congenital Nephrogenic Diabetes Insipidus and Regulation by Post-Translational Modifications and Protein-Protein Interactions"

_cells, 2020, doi:10.3390/cells9102172_

Round 1

Reviewer 1 Report

An interesting review reflecting the interests of the authors for the regulation of AQP2 by post translational modifications and the AQP2 protein/protein interactions.

page 6, line 90, ref 68 does not refer to AQP2.

page 8, line 127, AVPR2 receptors expressed in Xenopus oocytes? Ref?

recent reviews on NDI should be included:

    GENETICS IN ENDOCRINOLOGY Pathophysiology, diagnosis and treatment of familial nephrogenic diabetes insipidus. Bichet DG. Eur J Endocrinol. 2020 Aug;183(2):R29-R40       Pathophysiology, diagnosis and management of nephrogenic diabetes insipidus. Bockenhauer D, Bichet DG. Nat Rev Nephrol. 2015 Oct;11(10):576-88.

Reviewer 2 Report

General Comment: This is a nicely written review of AQP2 and mutations that cause congenital nephrogenic diabetes insipidus.

Major Comment: The title is a bit misleading.  The review really focuses on AQP2 mutations and mechanisms by which those mutations cause congenital nephrogenic diabetes insipidus.  While the review does cover AVPR2 mutations, this is a minor part of the review.  I would encourage the authors to change the title to better fit with the majority of the review.

Minor Comments:

page 10, line 182 - add "was" at the end of the line

page 13, line 239 - change recycling to recycles

page 15, lines 280 - what does "almost virtually appended" mean?

page 15, line 282 - add "the" before "Asn"

page 21, line 398 - change closes to close

Reviewer 3 Report

The authors review genetic and cell biologic aspects of NDI. Overall this is well-written. As a clinician, I will focus mainly on genetic and clinically related aspects.

The authors may want to discuss secondary inherited NDI and its potential pathophysiologic basis.

Because of the ambiguity of the term “mutation”, the aim in the genetics community is to use the neutral term “variant’ instead, which can then be further clarified by the adjectives “benign” or “pathogenic”.

In the summary, the authors mention that understanding the regulatory mechanism may hold promise for future management of the disease. This is an obvious standard sentence in any paper or grant application to emphasise clinical relevance. However, it would be much more interesting, if the review could be written under this aspect. How does the understanding of 14-3-3 or Hsc70 or of any of the other 139 interacting proteins specifically provide new therapeutic possibilities? This should be highlighted with each interacting partner discussed.

Specific comments:

  • Line 91: meant is “Intravascular” fluid?
  • Line 104: reference?
  • Line 110: reference?
  • Line 113: the last part of the sentence “and AQP2…” appears to be out of context with the following conclusion and seems top belong to the next paragraph
  • Line 121 ff: all these receptors are expressed in the principal cell? They are also GPCR? Or how do they affect AQP2?
  • Line 132: Mutations concentrate in the Golgi?
  • Line 153: how can AQP2 be unglygosylated AND high-mannose glycosylated? Better “or”?
  • Line 155 ff: repetitive from before?
  • Line 179: wording: “in patients from two…’? or just: “…In two families?”
  • Line 182: “WAS not retained…”
  • Line 178 ff: I am confused: the P262L variant is a “unique dominant mutation”, identified in recessive families and carriers are asymptomatic? That makes no sense?
  • Line 279 ff and 363ff: why the underlines?
  • Fig 2 and table 2: on what basis where the interacting proteins selected from the total of 139 interacting partners (see line 413)?
  • Table 1: legend missing?
